REGISTERED REPORT PROTOCOL

# A scoping review protocol on diagnostic strategies to detect occult malignancies in individuals with ischemic stroke

Jenneke Leentjens[1], Nicholas L. J. Chornenki[2], Janneke Spiegelenberg[1,3], Valentina Ly[4], Dariush Dowlatshahi[5], Deborah M. Siegal[5]*

1 Department of Internal Medicine & Radboud Institute for Health Sciences, Radboud University Medical Center, Nijmegen, The Netherlands, 2 Internal Medicine Residency Program, Queens University, Kingston, Canada, 3 Synapse Research Institute, Maastricht, The Netherlands, 4 Health Sciences Library, University of Ottawa, Ottawa, Canada, 5 Department of Medicine, University of Ottawa, Ottawa, Canada

* rcarr@toh.ca@debsiegal

## Abstract

### Background

Emerging data show an increased risk of ischemic stroke in patients with a new diagnosis of cancer. As the risk of stroke begins to increase 150 days before cancer is diagnosed, stroke may be the first clinical manifestation of undiagnosed cancer. About 6% of patients with cryptogenic ischemic stroke (unknown etiology after diagnostic evaluations) are diagnosed with cancer within one year. However, the optimal cancer screening strategy in this population is not known. We aim to conduct a scoping review of screening strategies for occult cancer in individuals with ischemic stroke.

### Methods

Electronic databases including MEDLINE (Ovid), EMBASE (Ovid), CINAHL (EBSCOhost) and Scopus will be systematically searched to identify articles that report on screening strategies for occult cancer in individuals with ischemic stroke. At least two investigators will independently perform two-stage study selection consisting of title/abstract screening and full-text review, followed by data extraction. Thereafter, a thematic analysis will be conducted to provide an overview of what diagnostic tests/strategies have been used, and their clinical utility in terms of positive and negative predictive value (when available).

### Conclusion

We anticipate that the findings of this scoping review will identify strategies used to detect occult cancer in individuals with ischemic stroke and summarize their clinical utility (if reported). Addressing this knowledge gap will help guide the development of future clinical trials on occult cancer screening patients with ischemic stroke.

**Funding:** This research received no specific grant from any funding agency in the public, commercial or not-for-profit sectors. NLJC and DMS are members of the Canadian Venous Thromboembolism Research Network (CanVECTOR); the Network received grant funding from the Canadian Institutes of Health Research (Funding Reference: CDT-142654). DMS is supported by a Tier 2 Canada Research Chair in Anticoagulant Treatment of Cardiovascular Disease. JL is supported by an ISTH research fellowship.

**Competing interests:** DMS received honoraria for educational presentations in the last 12 months (paid to institution) from BMS-Pfizer, Roche, and Servier. All other authors have declared that no competing interests exist,

## Introduction

Ischemic stroke is a leading cause of disability and mortality worldwide, affecting millions of people [1, 2]. Determining stroke etiology is key for optimal secondary stroke prevention, but remains unidentified in 10% to 40% of patients even after modern diagnostic workup (so-called cryptogenic stroke) [3, 4]. The current definition of cryptogenic stroke is based on the definition used in the trial of ORG 10172 in Acute Stroke Treatment (TOAST) [5]. Cryptogenic stroke (or "stroke of undetermined etiology" in TOAST terminology) is defined as an ischemic brain infarction not attributable to large-artery atherosclerosis, small artery disease, or cardioembolism despite a standard vascular, cardiac, and serologic evaluation (Table 1). Emerging data suggest that active cancer is an important underlying cause of ischemic stroke in individuals with cryptogenic stroke, with unique pathophysiology and treatment considerations [6]. Several large studies have demonstrated an increased risk of ischemic stroke in patients with new cancer diagnoses compared to matched controls [7, 8]. It is also increasingly recognized that stroke can precede cancer diagnosis by several months indicating that stroke may be the first manifestation of underlying cancer [9, 10]. Up to 10% of individuals with cryptogenic ischemic stroke are diagnosed with cancer within one year, which is similar to the incidence of occult cancer in individuals with unprovoked (no identifiable major risk factors) thromboembolism (VTE) [10–12]. However, compared to the extensive literature on cancer associated VTE, research on cancer associated stroke is more limited [11]. Although experts endorse consideration of underlying occult cancer as an etiology of cryptogenic stroke [6, 13, 14], specific recommendations are lacking and no advice is given in current guidelines [15–20]. Given the uncertainty about the extent and type of existing literature regarding screening for occult cancer in patients with ischemic stroke, we will map the available evidence in this scoping review in order to identify current knowledge gaps prior to undertaking a systematic review.

## Methods

This scoping review will be conducted according to the methodology proposed by Arksey and O'Malley [21, 22] which consists of nine stages as follows: 1) identify the research question(s); 2) develop the inclusion criteria; 3) describe the planned approach to evidence searching, selection, data extraction, and presentation; 4) search for the evidence; 5) select the evidence; 6) extract the evidence; 7) analyze the evidence; 8) present the results; 9) summarize the evidence in relation to the purpose of the review, making conclusions and noting any implications of the findings. This study is registered with the Open Science Framework (https://osf.io/3h95q). The results will be reported according to the Preferred Reporting Items for Systematic reviews and Meta-Analyses extension for Scoping Reviews (PRISMA-ScR).

### Research question

In patients with cryptogenic ischemic stroke, have occult cancer screening strategies been evaluated and, if so, what tests have been used and what is the clinical value in terms of positive and negative predictive value?

**Table 1. Routine workup according to TOAST criteria for stroke of undetermined etiology [13].**

| |
|---|
| Brain imaging by CT or MRI |
| Duplex ultrasound of the neck and transcranial doppler of the intracranial arteries or CT angiography of MR angiography of the neck and head arteries |
| Extensive cardiac workup (standard 12-lead electrocardiogram (ECG), at least 24-hour cardiac monitoring after stroke onset to detect subclinical atrial fibrillation, and transthoracic echocardiography) |
| Routine blood tests include blood glucose levels, complete blood count, International normalized ratio (INR), prothrombin time, activated partial thromboplastin time, and blood lipids |

## Inclusion and exclusion criteria

All articles that describe strategies for detecting occult cancers in adult patients with ischemic stroke (including, but not limited to, imaging tests, laboratory investigations, clinical assessments) will be assessed. Eligible articles include expert opinions, guidelines, narrative and systematic reviews, case-series and case-reports, conference abstracts, observational and randomized studies. Non-English reports, animal studies, studies limited to intracerebral hemorrhage, studies in patients with known cancers, studies on cardioembolic ischemic stroke (including fibroelastoma, atrial myxoma, and non-bacterial thrombotic endocarditis), and studies without details about screening strategies will be excluded. The reference lists of articles selected for full-text review will also be reviewed for potentially eligible studies that were not captured by the search.

## Search strategy and information sources

The search strategy was developed by a health sciences librarian (VL) and peer-reviewed by an independent health sciences librarian from the University of Ottawa (Tables 2–5) [23]. MED-LINE (Ovid), Embase (Ovid), CINAHL (EBSCO), and Scopus were searched with no limits to language or publication date using appropriate subject headings and keywords (see S1 Checklist for full search details). The main search concepts comprise of terms related to ischemic stroke and cancer screening. Search results were exported to Covidence (Melbourne, Australia) and duplicates were eliminated using the platform's duplicate identification feature.

## Article selection

We will use a two-stage study selection of title and abstract screening, followed by full-text review, to be completed by two independent reviewers [24]. Disagreements will be resolved by discussion, or involvement of a third reviewer if consensus cannot be reached. A PRISMA flow diagram will be used to summarize the process of article selection.

**Table 2. Medline (Ovid) search strategy.**

| # | Searches | Results |
|---|---|---|
| 1 | neoplasms/ and (screen* or test* or diagnos* or detect* or predict* or identif* or scan* or biomarker* or marker* or metabolite* or biops*).ti,ab,kf. | 169280 |
| 2 | early detection of cancer/ | 34210 |
| 3 | biomarkers, tumor/ | 172208 |
| 4 | ((oncolog* or cancer* or carcinoma* or tumor* or tumour* or neoplasm* or metasta* or malignan* or hodgkin* or nonhodgkin* or adenocarcinoma* or leukemi* or leukaemi* or lymphoma* or sarcoma* or myeloma* or melanoma*) adj5 (screen* or test* or diagnos* or detect* or predict* or identif* or scan* or biomarker* or marker* or metabolite* or biops*)).ti,ab,kf. | 872000 |
| 5 | or/1-4 | 1049172 |
| 6 | exp ischemic stroke/ | 7015 |
| 7 | stroke/ and (ischemi* or ischaemi* or cryptogenic* or wake* or awak* or embol* or cardioembol* or thrombo* or lacunar).ti,ab,kf. | 57085 |
| 8 | ((ischemi* or ischaemi* or cryptogenic* or wake* or awak* or embol* or cardioembol* or thrombo*) adj3 (stroke* or cerebr* vascul* or cerebrovascul* or CVA or CVAs or apoplex*)).ti,ab,kf. | 90997 |
| 9 | ((ischemi* or ischaemi* or cryptogenic* or wake* or awak* or embol* or cardioembol* or thrombo*) adj3 (brain adj2 (insult* or accident* or attack*))).ti,ab,kf. | 249 |
| 10 | (lacunar adj2 (stroke* or syndrome* or infarct*)).ti,ab,kf. | 4133 |
| 11 | or/6-10 | 106634 |
| 12 | 5 and 11 | 930 |

**Table 3. Embase (Ovid) search strategy.**

| # | Searches | Results |
|---|----------|---------|
| 1 | neoplasm/ and (screen* or test* or diagnos* or detect* or predict* or identif* or scan* or biomarker* or marker* or metabolite* or biops*).ti,ab,kf. | 287402 |
| 2 | cancer screening/ | 89245 |
| 3 | tumor marker/ | 93722 |
| 4 | ((oncolog* or cancer* or carcinoma* or tumor* or tumour* or neoplasm* or metasta* or malignan* or hodgkin* or nonhodgkin* or adenocarcinoma* or leukemi* or leukaemi* or lymphoma* or sarcoma* or myeloma* or melanoma*) adj5 (screen* or test* or diagnos* or detect* or predict* or identif* or scan* or biomarker* or marker* or metabolite* or biops*)).ti,ab,kf. | 1317283 |
| 5 | or/1-4 | 1533724 |
| 6 | exp ischemic stroke/ | 14021 |
| 7 | cerebrovascular accident/ and (ischemi* or ischaemi* or cryptogenic* or wake* or awak* or embol* or cardioembol* or thrombo* or lacunar).ti,ab,kf. | 90841 |
| 8 | ((ischemi* or ischaemi* or cryptogenic* or wake* or awak* or embol* or cardioembol* or thrombo*) adj3 (stroke* or cerebr* vascul* or cerebrovascul* or CVA or CVAs or apoplex*)).ti,ab,kf. | 153077 |
| 9 | ((ischemi* or ischaemi* or cryptogenic* or wake* or awak* or embol* or cardioembol* or thrombo*) adj3 (brain adj2 (insult* or accident* or attack*))).ti,ab,kf. | 373 |
| 10 | (lacunar adj2 (stroke* or syndrome* or infarct*)).ti,ab,kf. | 6973 |
| 11 | or/6-10 | 195402 |
| 12 | 5 and 11 | 2772 |

**Table 4. CINAHL (EBSCO) search strategy.**

| # | Searches | Results |
|---|----------|---------|
| S1 | (MH "Neoplasms") | 89,751 |
| S2 | TI ((screen* or test* or diagnos* or detect* or predict* or identif* or scan* or biomarker* or marker* or metabolite* or biops*)) OR AB ((screen* or test* or diagnos* or detect* or predict* or identif* or scan* or biomarker* or marker* or metabolite* or biops*)) | 2,239,282 |
| S3 | S1 AND S2 | 33,284 |
| S4 | (MH "Early Detection of Cancer") | 11,657 |
| S5 | (MH "Tumor Markers, Biological") | 16,863 |
| S6 | TI (((oncolog* or cancer* or carcinoma* or tumor* or tumour* or neoplasm* or metasta* or malignan* or hodgkin* or nonhodgkin* or adenocarcinoma* or leukemi* or leukaemi* or lymphoma* or sarcoma* or myeloma* or melanoma*) N5 (screen* or test* or diagnos* or detect* or predict* or identif* or scan* or biomarker* or marker* or metabolite* or biops*)) OR AB (((oncolog* or cancer* or carcinoma* or tumor* or tumour* or neoplasm* or metasta* or malignan* or hodgkin* or nonhodgkin* or adenocarcinoma* or leukemi* or leukaemi* or lymphoma* or sarcoma* or myeloma* or melanoma*) N5 (screen* or test* or diagnos* or detect* or predict* or identif* or scan* or biomarker* or marker* or metabolite* or biops*))) | 177,839 |
| S7 | S3 OR S4 OR S5 OR S6 | 204,995 |
| S8 | (MH "Ischemic Stroke+") | 642 |
| S9 | (MH "Stroke") | 76,233 |
| S10 | TI ((ischemi* or ischaemi* or cryptogenic* or wake* or awak* or embol* or cardioembol* or thrombo* or lacunar)) OR AB ((ischemi* or ischaemi* or cryptogenic* or wake* or awak* or embol* or cardioembol* or thrombo* or lacunar)) | 178,863 |
| S11 | S9 AND S10 | 22,165 |
| S12 | TI (((ischemi* or ischaemi* or cryptogenic* or wake* or awak* or embol* or cardioembol* or thrombo*) N3 (stroke* or "cerebr* vascul*" or cerebrovascul* or CVA or CVAs or apoplex*))) OR AB (((ischemi* or ischaemi* or cryptogenic* or wake* or awak* or embol* or cardioembol* or thrombo*) N3 (stroke* or "cerebr* vascul*" or cerebrovascul* or CVA or CVAs or apoplex*))) | 30,904 |
| S13 | TI (((ischemi* or ischaemi* or cryptogenic* or wake* or awak* or embol* or cardioembol* or thrombo*) N3 (brain N2 (insult* or accident* or attack*)))) OR AB (((ischemi* or ischaemi* or cryptogenic* or wake* or awak* or embol* or cardioembol* or thrombo*) N3 (brain N2 (insult* or accident* or attack*)))) | 63 |

*(Continued)*

**Table 4.** (Continued)

| # | Searches | Results |
|---|----------|---------|
| S14 | (MH "Stroke, Lacunar") | 293 |
| S15 | TI ((lacunar N2 (stroke* or syndrome* or infarct*))) OR AB ((lacunar N2 (stroke* or syndrome* or infarct*))) | 1,286 |
| S16 | S8 OR S11 OR S12 OR S13 OR S14 OR S15 | 35,765 |
| S17 | S7 AND S16 | 318 |

## Data extraction and management

For studies that meet the inclusion criteria, relevant data will be extracted and captured using a standardized data extraction form. Specific data to be collected are shown in Table 6.

## Conclusion

The objective of the scoping review described in this protocol is to summarize the available evidence on screening strategies to detect occult cancer in individuals with ischemic stroke. We anticipate that this review will summarize the type of tests and/or strategies that are used in clinical practice and provide preliminary evidence about their utility, but there will be heterogeneity due to existing knowledge gaps about the optimal strategy. Information gathered from this review will also help to guide the development of future clinical trials on screening strategies for occult cancer in individuals with ischemic stroke.

**Table 5. Scopus search strategy.**

| # | Searches | Results |
|---|----------|---------|
| 1 | TITLE-ABS-KEY ((oncolog* OR cancer* OR carcinoma* OR tumor* OR tumour* OR neoplasm* OR metasta* OR malignan* OR hodgkin* OR nonhodgkin* OR adenocarcinoma* OR leukemi* OR leukaemi* OR lymphoma* OR sarcoma* OR myeloma* OR melanoma*) W/5 (screen* OR test* OR diagnos* OR detect* OR predict* OR identif* OR scan* OR biomarker* OR marker* OR metabolite* OR biops*)) | 1.324.376 |
| 2 | TITLE-ABS-KEY ((ischemi* OR ischaemi* OR cryptogenic* OR wake* OR awak* OR embol* OR cardioembol* OR thrombo*) W/3 (stroke* OR "cerebr* vascul*" OR cerebrovascul* OR cva OR cvas OR apoplex*)) | 108.780 |
| 3 | TITLE-ABS-KEY ((ischemi* OR ischaemi* OR cryptogenic* OR wake* OR awak* OR embol* OR cardioembol* OR thrombo*) W/3 (brain W/2 (insult* OR accident* OR attack*))) | 848 |
| 4 | TITLE-ABS-KEY (lacunar W/2 (stroke* OR syndrome* OR infarct*)) | 6.316 |
| 5 | (TITLE-ABS-KEY ((ischemi* OR ischaemi* OR cryptogenic* OR wake* OR awak* OR embol* OR cardioembol* OR thrombo*) W/3 (stroke* OR "cerebr* vascul*" OR cerebrovascul* OR cva OR cvas OR apoplex*))) OR (TITLE-ABS-KEY ((ischemi* OR ischaemi* OR cryptogenic* OR wake* OR awak* OR embol* OR cardioembol* OR thrombo*) W/3 (brain W/2 (insult* OR accident* OR attack*)))) OR (TITLE-ABS-KEY (lacunar W/2 (stroke* OR syndrome* OR infarct*))) | 113.600 |
| 6 | (TITLE-ABS-KEY ((oncolog* OR cancer* OR carcinoma* OR tumor* OR tumour* OR neoplasm* OR metasta* OR malignan* OR hodgkin* OR nonhodgkin* OR adenocarcinoma* OR leukemi* OR leukaemi* OR lymphoma* OR sarcoma* OR myeloma* OR melanoma*) W/5 (screen* OR test* OR diagnos* OR detect* OR predict* OR identif* OR scan* OR biomarker* OR marker* OR metabolite* OR biops*))) AND ((TITLE-ABS-KEY ((ischemi* OR ischaemi* OR cryptogenic* OR wake* OR awak* OR embol* OR cardioembol* OR thrombo*) W/3 (stroke* OR "cerebr* vascul*" OR cerebrovascul* OR cva OR cvas OR apoplex*))) OR (TITLE-ABS-KEY ((ischemi* OR ischaemi* OR cryptogenic* OR wake* OR awak* OR embol* OR cardioembol* OR thrombo*) W/3 (brain W/2 (insult* OR accident* OR attack*)))) OR (TITLE-ABS-KEY (lacunar W/2 (stroke* OR syndrome* OR infarct*)))) | 979 |

**Table 6. Data included in the standardized data extraction form.**

| |
|---|
| First author(s) |
| Year of publication |
| Origin/country of origin (where the study was published or conducted) |
| Objective(s)Study design |
| Duration of follow-up* |
| Study population (inclusion and exclusion criteria)* |
| Sample size* |
| Intervention type/duration, comparator* |
| Outcome measures *<br>  • Definition of cancer diagnosis |
| Key findings: screening strategies/diagnostic tests used (including but not limited to laboratory tests, imaging investigations, clinical characteristics), PPV/NPV* |

*if applicable.

## Supporting information

**S1 Checklist. PRISMA-P (Preferred Reporting Items for Systematic review and Meta-Analysis Protocols) 2015 checklist: Recommended items to address in a systematic review protocol*.**
(DOC)

## Acknowledgments

We thank Victoria Cole, BScN, MScN, MIS (Health Sciences Library, University of Ottawa) for peer review of the MEDLINE search strategy.

## Author Contributions

**Conceptualization:** Jenneke Leentjens, Dariush Dowlatshahi, Deborah M. Siegal.

**Data curation:** Valentina Ly.

**Formal analysis:** Jenneke Leentjens.

**Methodology:** Jenneke Leentjens, Nicholas L. J. Chornenki, Valentina Ly, Deborah M. Siegal.

**Project administration:** Jenneke Leentjens, Janneke Spiegelenberg, Valentina Ly.

**Software:** Nicholas L. J. Chornenki, Janneke Spiegelenberg, Valentina Ly.

**Supervision:** Dariush Dowlatshahi, Deborah M. Siegal.

**Writing – original draft:** Jenneke Leentjens.

**Writing – review & editing:** Nicholas L. J. Chornenki, Janneke Spiegelenberg, Valentina Ly, Dariush Dowlatshahi, Deborah M. Siegal.

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
