## [Decision Letter · Decision Letter 0]

21 Dec 2022

PONE-D-22-25756A scoping review protocol on diagnostic strategies to detect occult malignancies in individuals with ischemic strokePLOS ONE

Dear Dr. Leentjens,

Thank you for submitting your manuscript to PLOS ONE. After careful consideration, we feel that it has merit but does not fully meet PLOS ONE’s publication criteria as it currently stands. Therefore, we invite you to submit a revised version of the manuscript that addresses the points raised during the review process.

We look forward to receiving your revised manuscript.

Kind regards,

Fateen Ata, MD

Academic Editor

PLOS ONE

Journal Requirements:

Additional Editor Comments:

While it is an interesting review topic, the reviewers have suggested some integral revisions to be made, after which the protocol can be reviewed again.

Reviewers' comments:

Reviewer's Responses to Questions

**Comments to the Author**

1. Does the manuscript provide a valid rationale for the proposed study, with clearly identified and justified research questions?

Reviewer #1: Partly

Reviewer #2: Yes

Reviewer #3: Yes

Reviewer #4: Yes

2. Is the protocol technically sound and planned in a manner that will lead to a meaningful outcome and allow testing the stated hypotheses?

Reviewer #1: Yes

Reviewer #2: Yes

Reviewer #3: Yes

Reviewer #4: Yes

3. Is the methodology feasible and described in sufficient detail to allow the work to be replicable?

Reviewer #1: Yes

Reviewer #2: Yes

Reviewer #3: Yes

Reviewer #4: Yes

4. Have the authors described where all data underlying the findings will be made available when the study is complete?

Reviewer #1: Yes

Reviewer #2: Yes

Reviewer #3: No

Reviewer #4: Yes

5. Is the manuscript presented in an intelligible fashion and written in standard English?

Reviewer #1: Yes

Reviewer #2: Yes

Reviewer #3: Yes

Reviewer #4: Yes

6. Review Comments to the Author

You may also provide optional suggestions and comments to authors that they might find helpful in planning their study.

Reviewer #1: 1. It is unclear why the authors chose to pursue a scoping review instead of a systematic review for this topic. At least some explanation is necessary.

2. There is a very recent review published on the same topic in 2022 (see: pubmed.ncbi.nlm.nih.gov/36192850). This should be referenced and the rationale for the present review better justified.

Reviewer #2: Dear editor,

I reviewed the article received by Leentjens et all on scoping review on cancer as possible etiology of infarctive stroke. The protocol is well written and the strategies part in the tables in elucidated well.

I have just one question for the authors

1. Have figured out any statistical analysis plans for the data? If yes what kind of data would the team extract and how are they planning on analysing it?

2. While its clearly a scoping review, I am interested to know how have the authors planned on presenting the findings from expert opinions, narratives and guidelines which they listed as part of inclusion criteria. Will they present descriptively side by side with data analysis?

If there is a plan to run statistical analysis, the authors need to elaborate on that clearly. Otherwisethere is any need to update the manuscript.

I wish the authors best of luck with their submission and the review

Reviewer #3: This manuscript sheds light on the long neglected topic of occult cancer in ischaemic stroke patients. The knowledge gap regarding the screening strategy end efforts to diagnose, or even more important exclude the possibility of underlying cancer, is clinically highly relevant. I am looking forward to see the results of your work.

I have some minor comments to the abstract: I did not found proper reference for the facts you present in the Background *''Up to 10% of individuals with cryptogenic ischemic stroke are diagnosed with cancer within one year, which is similar to the incidence of occult cancer in individuals with unprovoked (no identifiable risk factors) thromboembolism (VTE) (10, 11). '' Ref 10 and 11 regard VTE patients and not ischaemic stroke, most of them due to arterial source.

My other comment is about the methodology, could you please elaborate the method part, regarding type of the review you chose.

Reviewer #4: Thank you for the invitation to review the manuscript which is research about the occult malignancies in patients with ischemic stroke. In my opinion this scientific protocol is laborious and helpful work in this area. I hope that studies in the subject will continue.

Some of the literature is older than 5 years but there are articles from 2021/22 in the manuscript so the authors also base on new researches.

Minor comments

The reference (#7) is a report of meta-analysis, so if the authors would like to cite it as is, they should change the description in the text from “several studies” to "meta-analysis". If the authors would like to keep the description, the reviewer recommend citing the study by Navi et al (PMID: 28818202) and Mulder et al (PMID: 34396325).

7. PLOS authors have the option to publish the peer review history of their article (what does this mean?). If published, this will include your full peer review and any attached files.

Reviewer #1: No

Reviewer #2: No

Reviewer #3: **Yes: **Barbara Ratajczak-Tretel

Reviewer #4: No

---

## [Author Response · Author response to Decision Letter 0]

2 Mar 2023

Dear editor,

I reviewed the article received by Leentjens et all on scoping review on cancer as possible etiology of infarctive stroke. The protocol is well written and the strategies part in the tables in elucidated well.

I have just one question for the authors

1. Have figured out any statistical analysis plans for the data? If yes what kind of data would the team extract and how are they planning on analysing it?

R: The objective of our review is to summarize the evidence from available studies . We will use descriptive statistics to summarize our findings including mean with standard deviation, or median with interquartile range for continuous variables, and proportions for categorical variables, where appropriate. We will calculate the positive/negative predictive value of a diagnostic test, when available but will not perform comparative analyses between studies, nor will we pool the results in meta-analysis. 

2. While it is clearly a scoping review, I am interested to know how have the authors planned on presenting the findings from expert opinions, narratives and guidelines which they listed as part of inclusion criteria. Will they present descriptively side by side with data analysis?

R: We will summarize the recommendations from narrative reviews and guidelines in separately from other studies. The type of article (narrative review/guideline) will be stated in the “study design” section and their recommendations in the “key-findings” section. For narrative reviews/guidelines all other items in the chart will be scored as “not applicable”. 

3. If there is a plan to run statistical analysis, the authors need to elaborate on that clearly. Otherwise there is no need to update the manuscript. 

R: we do not plan to run perform comparative analysis nor will we pool the results in meta-analysis.

---

## [Decision Letter · Decision Letter 1]

22 Mar 2023

PONE-D-22-25756R1A scoping review protocol on diagnostic strategies to detect occult malignancies in individuals with ischemic strokePLOS ONE

Dear Dr.  Leentjens,

Thank you for submitting your manuscript to PLOS ONE. After careful consideration, we feel that it has merit but does not fully meet PLOS ONE’s publication criteria as it currently stands. Therefore, we invite you to submit a revised version of the manuscript that addresses the points raised during the review process.

ACADEMIC EDITOR:Dear Authors, In the revised manuscript you have commented on only 1 reviewer's comments, to proceed further you need to address the comments raised by all reviewers.

We look forward to receiving your revised manuscript.

Kind regards,

Fateen Ata, MD

Academic Editor

PLOS ONE

Journal Requirements:

Additional Editor Comments (if provided):

Dear Authors, In the revised manuscript you have commented on only 1 reviewer's comments, to proceed further you need to address the comments raised by all reviewers.

Reviewers' comments:

Reviewer's Responses to Questions

**Comments to the Author**

1. Does the manuscript provide a valid rationale for the proposed study, with clearly identified and justified research questions?

Reviewer #1: Partly

Reviewer #4: Yes

2. Is the protocol technically sound and planned in a manner that will lead to a meaningful outcome and allow testing the stated hypotheses?

Reviewer #1: Yes

Reviewer #4: Yes

3. Is the methodology feasible and described in sufficient detail to allow the work to be replicable?

Reviewer #1: Yes

Reviewer #4: Yes

4. Have the authors described where all data underlying the findings will be made available when the study is complete?

Reviewer #1: Yes

Reviewer #4: Yes

5. Is the manuscript presented in an intelligible fashion and written in standard English?

Reviewer #1: Yes

Reviewer #4: Yes

6. Review Comments to the Author

You may also provide optional suggestions and comments to authors that they might find helpful in planning their study.

Reviewer #1: Thank you for the revisions. The manuscript is acceptable. I have no further comments to the authors.

Reviewer #4: The reviewer has reviewed the revised manuscript made by the authors. Unfortunately, the authors' responses to the Reviewers' comments are inadequate.

7. PLOS authors have the option to publish the peer review history of their article (what does this mean?). If published, this will include your full peer review and any attached files.

Reviewer #1: No

Reviewer #4: No

---

## [Author Response · Author response to Decision Letter 1]

19 Jun 2023

Reviewer 1:

1. It is unclear why the authors chose to pursue a scoping review instead of a systematic review for this topic. At least some explanation is necessary.

R: Thank you for the opportunity to clarify the rationale for a scoping review on the topic of cancer screening after cryptogenic ischemic stroke.. 

Action: we added “Given the uncertainty about the extent and type of existing literature regarding screening for occult cancer in patients with ischemic stroke and likely heterogeneity in the type of tests evaluated and settings in which testing is conducted, we will map the available evidence in this scoping review in order to identify current knowledge gaps prior to undertaking a systematic review.” to the introduction paragraph. 

2. There is a very recent review published on the same topic in 2022 (see: pubmed.ncbi.nlm.nih.gov/36192850). This should be referenced and the rationale for the present review better justified.

R: There is significant interest in understanding the relationship between occult cancer and unexplained thrombosis (both venous and arterial) and whether enhanced detection can improve prognosis. The citation provided addresses screening for occult cancer after unprovoked venous thromboembolism. Although there may be some overlap for these cardiovascular conditions, they are distinct vascular diseases affecting different patient populations with different etiologies and natural histories such that the data cannot be generalized. 

Reviewer 2:

Dear editor,

I reviewed the article received by Leentjens et all on scoping review on cancer as possible etiology of infarctive stroke. The protocol is well written and the strategies part in the tables in elucidated well.

I have just one question for the authors

1. Have figured out any statistical analysis plans for the data? If yes what kind of data would the team extract and how are they planning on analysing it?

R: We will not perform any statistical analysis apart from the calculation of the positive/negative predictive value of a diagnostic test, when available. The design of our scoping review is purely descriptive. We will chart the available studies but do not intend to perform qualitative comparisons between studies.

2. While its clearly a scoping review, I am interested to know how have the authors planned on presenting the findings from expert opinions, narratives and guidelines which they listed as part of inclusion criteria. Will they present descriptively side by side with data analysis?

R: We will include the recommendations from narrative reviews and guidelines in the standardized data extraction form, side to side to other relevant studies. The type of article (narrative review/guideline) will be stated in the “study design” section and their recommendations in the “key-findings” section. For narrative reviews/guidelines all other items in the chart will be scored as “not applicable”. 

Reviewer 3:

This manuscript sheds light on the long neglected topic of occult cancer in ischaemic stroke patients. The knowledge gap regarding the screening strategy end efforts to diagnose, or even more important exclude the possibility of underlying cancer, is clinically highly relevant. I am looking forward to see the results of your work.

1. I have some minor comments to the abstract: I did not found proper reference for the facts you present in the Background *''Up to 10% of individuals with cryptogenic ischemic stroke are diagnosed with cancer within one year, which is similar to the incidence of occult cancer in individuals with unprovoked (no identifiable risk factors) thromboembolism (VTE) (10, 11). '' Ref 10 and 11 regard VTE patients and not ischaemic stroke, most of them due to arterial source.

R: we added PMID 31231302 as relevant reference regarding ischemic stroke. 

2. My other comment is about the methodology, could you please elaborate the method part, regarding type of the review you chose.

R: Thank you for the opportunity to clarify the rationale for scoping review. 

Action: we added “Given the uncertainty about the extent and type of existing literature regarding screening for occult cancer in patients with ischemic stroke stroke and likely heterogeneity in the type of tests evaluated and settings in which testing is conducted, we will map the available evidence in this scoping review in order to identify current knowledge gaps prior to undertaking a systematic review.” to the introduction paragraph. 

Reviewer 4:

Thank you for the invitation to review the manuscript which is research about the occult malignancies in patients with ischemic stroke. In my opinion this scientific protocol is laborious and helpful work in this area. I hope that studies in the subject will continue.

Some of the literature is older than 5 years but there are articles from 2021/22 in the manuscript so the authors also base on new researches.

Minor comments

The reference (#7) is a report of meta-analysis, so if the authors would like to cite it as is, they should change the description in the text from “several studies” to "meta-analysis". If the authors would like to keep the description, the reviewer recommend citing the study by Navi et al (PMID: 28818202) and 

R: we thank the reviewer for this suggestion and changed the manuscript accordingly (and added PMID: 34396325, and 28818202 instead of the meta-analysis).

---

## [Decision Letter · Decision Letter 2]

11 Jul 2023

A scoping review protocol on diagnostic strategies to detect occult malignancies in individuals with ischemic stroke

PONE-D-22-25756R2

Dear Dr. Leentjens,

We’re pleased to inform you that your manuscript has been judged scientifically suitable for publication and will be formally accepted for publication once it meets all outstanding technical requirements.

Kind regards,

Fateen Ata, MD

Academic Editor

PLOS ONE

Additional Editor Comments (optional):

Reviewers' comments:

Reviewer's Responses to Questions

**Comments to the Author**

1. Does the manuscript provide a valid rationale for the proposed study, with clearly identified and justified research questions?

Reviewer #4: Yes

2. Is the protocol technically sound and planned in a manner that will lead to a meaningful outcome and allow testing the stated hypotheses?

Reviewer #4: Yes

3. Is the methodology feasible and described in sufficient detail to allow the work to be replicable?

Reviewer #4: Yes

4. Have the authors described where all data underlying the findings will be made available when the study is complete?

Reviewer #4: Yes

5. Is the manuscript presented in an intelligible fashion and written in standard English?

Reviewer #4: Yes

6. Review Comments to the Author

You may also provide optional suggestions and comments to authors that they might find helpful in planning their study.

Reviewer #4: The authors have addressed the reviewers' comments adequately. I have no additional comments to provide.

7. PLOS authors have the option to publish the peer review history of their article (what does this mean?). If published, this will include your full peer review and any attached files.

Reviewer #4: No

---

## [Editor Report · Acceptance letter]

14 Jul 2023

PONE-D-22-25756R2 

A scoping review protocol on diagnostic strategies to detect occult malignancies in individuals with ischemic stroke. 

Dear Dr. Leentjens:

I'm pleased to inform you that your manuscript has been deemed suitable for publication in PLOS ONE. Congratulations! Your manuscript is now with our production department. 

Kind regards, 

on behalf of

Dr. Fateen Ata 

Academic Editor

PLOS ONE